# Predicting optimal deep brain stimulation parameters for Parkinson's disease using functional MRI and machine learning

Alexandre Boutet [1,2,8], Radhika Madhavan[3,8], Gavin J. B. Elias[2], Suresh E. Joel[4], Robert Gramer[2], Manish Ranjan [2], Vijayashankar Paramanandam[5], David Xu[2], Jurgen Germann [2], Aaron Loh[2], Suneil K. Kalia[2], Mojgan Hodaie [2], Bryan Li[1], Sreeram Prasad[5], Ailish Coblentz[1], Renato P. Munhoz[5], Jeffrey Ashe[3], Walter Kucharczyk[1], Alfonso Fasano [5,6,7] & Andres M. Lozano[2,6,7 ✉]

Commonly used for Parkinson's disease (PD), deep brain stimulation (DBS) produces marked clinical benefits when optimized. However, assessing the large number of possible stimulation settings (i.e., programming) requires numerous clinic visits. Here, we examine whether functional magnetic resonance imaging (fMRI) can be used to predict optimal stimulation settings for individual patients. We analyze 3 T fMRI data prospectively acquired as part of an observational trial in 67 PD patients using optimal and non-optimal stimulation settings. Clinically optimal stimulation produces a characteristic fMRI brain response pattern marked by preferential engagement of the motor circuit. Then, we build a machine learning model predicting optimal vs. non-optimal settings using the fMRI patterns of 39 PD patients with a priori clinically optimized DBS (88% accuracy). The model predicts optimal stimulation settings in unseen datasets: a priori clinically optimized and stimulation-naïve PD patients. We propose that fMRI brain responses to DBS stimulation in PD patients could represent an objective biomarker of clinical response. Upon further validation with additional studies, these findings may open the door to functional imaging-assisted DBS programming.

[1] Joint Department of Medical Imaging, University of Toronto, Toronto, Canada. [2] Division of Neurosurgery, Department of Surgery, University Health Network and University of Toronto, Toronto, ON, Canada. [3] GE Global Research Center, Niskayuna, NY, USA. [4] GE Healthcare, Bangalore, India. [5] Edmond J. Safra Program in Parkinson's Disease, Morton and Gloria Shulman Movement Disorders Clinic, Toronto Western Hospital, UHN, Division of Neurology, University of Toronto, Toronto, ON, Canada. [6] Krembil Brain Institute, Toronto, ON, Canada. [7] Center for Advancing Neurotechnological Innovation to Application (CRANIA), Toronto, ON, Canada. [8] These authors contributed equally: Alexandre Boutet, Radhika Madhavan. ✉email: lozano@uhnresearch.ca

Deep brain stimulation (DBS) has become a standard of care therapy for movement disorders, particularly Parkinson's disease (PD), essential tremor and dystonia, and is being investigated in psychiatric and cognitive disorders including major depressive disorder and Alzheimer's disease[1,2]. DBS involves placing an electrode to deliver electrical stimulation within a dysfunctional neural circuit to suppress aberrant activity and/or drive an underactive network. Despite its recognized benefits, the therapeutic mechanism of action of DBS remains incompletely understood[1].

The subthalamic nucleus (STN), an integral hub in the motor circuit, is the most common target in PD-DBS[3]. The success of DBS is critically dependent on delivering the appropriate dose of stimulation at the best location within the target region. DBS programming, the process of individually titrating the dose of electrical stimulation delivered to achieve maximal clinical benefits, remains largely a trial-and-error process predicated on immediate clinical observations and neurologist experience[4,5]. Some clinical features respond rapidly to electrical stimulation in PD-DBS, for example, rigidity and, less predictably, tremor. For other impairments, including bradykinesia, abnormal posture, and gait difficulties, where there can be slow and progressive benefits but also deleterious effects, empirical programming poses a significant challenge[4]. Beyond PD, programming is particularly difficult in patients with DBS for indications such as dystonia, depression, and Alzheimer's disease, in which the response to DBS typically occurs in a delayed fashion and may even be clinically occult for weeks to months following parameter adjustment. In each case, DBS programming requires multiple clinic visits (typically to tertiary health centers) to test the vast number of possible parameters and discover the setting that produces the greatest symptomatic relief with the least side-effects[4]. This process imposes significant time and financial stress upon patients and healthcare systems[6]. Hence, there is a need for a physiological marker that can rapidly and accurately predict clinical response to DBS parameters and improve the efficiency and lessen the burden of current programming practices[4].

Advances in neuroimaging techniques have furthered our understanding of the physiological effects of DBS on the activity of brain circuits (Supplementary Table 1). Since MRI in patients with DBS is subject to strict safety guidelines[7], studies have leveraged normative connectomes to retrospectively investigate brain regions and networks whose modulation is associated with clinical benefits[8]. Prospective functional magnetic resonance imaging (fMRI) acquisition in this patient population has largely been limited to studies using suboptimal MRI hardware due to safety concerns[7]. However, recent advances have established safety and feasibility of using a number of MRI sequences in patients with DBS[7,9] and have enabled a more detailed examination of the physiologic consequences of the application of DBS on specific brain circuits. fMRI is now being studied to probe the consequences of stimulation on brain networks[10–13], but it has not so far been used to predict optimal DBS stimulation parameters nor to directly enhance DBS's therapeutic potential.

In this work, we show that prospective fMRI data can identify brain activity patterns associated with clinical benefits in PD patients, serving as a biomarker of DBS efficacy. We use fMRI to (1) identify a reproducible pattern of brain response to optimal DBS stimulation and (2) predict optimal DBS settings on the basis of these brain response patterns with a machine learning (ML) algorithm. This algorithm was trained on already optimized PD patients and tested on two new datasets: an a priori clinically defined stimulation-optimized PD patient group and a stimulation-naïve PD patient cohort.

## Results

Building on prior publications describing the safety and feasibility of MRI in DBS patients[7,9,14], 3 T fMRI data were prospectively acquired over the course of 203 fMRI sessions ($n = 67$ PD-DBS patients, Fig. 1, Table 1). Since STN is the most common target for DBS in the management of PD, we primarily recruited STN-DBS patients ($n = 62$). We also included patients with internal globus pallidus (GPi) DBS ($n = 5$), which is a second commonly used stimulation location, to assess whether different PD-DBS targets could also contribute to the ML model (Table 1). Each session was 6.5 min in duration and employed a 30 s DBS-ON/ OFF cycling paradigm repeated six times in which unilateral left DBS stimulation was delivered at patient-specific, clinically defined optimal and non-optimal contacts or voltages (Fig. 1C). As previously reported[15], this was done to differentiate between the unilateral and contralateral BOLD signal changes, as well as to attempt to mimic DBS programming, which usually entails evaluating one electrode at a time. Acquired fMRI data were preprocessed using an established pipeline that performed motion and slice timing correction (Fig. 2). Blood-oxygen-level-dependent (BOLD) signal was extracted from 16 motor and non-motor regions-of-interests (ROIs) determined a priori based on existing PET and SPECT literature[16–19] and our experience with adverse effects (e.g., speech issues and visual disturbances) with non-optimal settings during DBS fMRI[20]. Given that fMRI studies have been uncommonly performed due to safety concerns, PET and SPECT have largely informed our ROIs choices. The absolute $t$-values (BOLD changes) were normalized by mean positive $t$-values in areas presumed to be involved in non-optimal stimulation. This was done to compare $t$-values of BOLD-response DBS-ON vs. DBS-OFF of each ROI across patients and to account for adverse effects—a key consideration given that the aim of DBS programming is to maximize motor benefits while minimizing adverse effects. Normalized BOLD changes (features) from 39 a priori clinically optimized patients ($n = 35$ STN-DBS and $n = 4$ GPi-DBS) and their associated binary labeling (optimal vs. non-optimal) were used as input to train the ML model (Fig. 2, Table 1). Clinically optimal DBS settings were obtained using published algorithms[4,5]. Subsequently, two unseen fMRI datasets ($n = 9$ for each dataset)—acquired with different active contacts or voltages—were fed into the trained ML model for validation purposes. The model's ability to determine whether a DBS setting was optimal or non-optimal according to the corresponding fMRI pattern was assessed (Fig. 2).

**Typical fMRI activation pattern with optimal stimulation**. The fMRI BOLD response maps and electrode locations in individual PD patients with the left STN-DBS electrode turned ON at clinically optimal and non-optimal contact settings (Fig. 3A, Supplementary Fig. S1A, B) and optimal and non-optimal (i.e., subtherapeutic and supratherapeutic) voltage settings (Fig. 3B, Supplementary Fig. S1C, D) are shown. Optimal left-sided STN-DBS stimulation (i.e., contact and voltage) produced significant BOLD signal changes in the motor circuit, including an increased signal in the left (ipsilateral) thalamus and decreased signal in the left (ipsilateral) primary motor cortex and right (contralateral) anterior cerebellum (Fig. 3A, B). Stimulation at non-optimal contacts (±3 or 6 mm center-to-center distance from optimal contact (mapped to 0 mm)) generated a diminished magnitude BOLD response in the primary motor cortex with associated BOLD signal increases in non-motor regions (e.g., visual cortex) (Fig. 3A). When using the optimal stimulation contact, decreasing stimulation intensity from optimal to low (subtherapeutic) voltage stimulation triggered a decrease in magnitude of the BOLD changes but maintained the topographic pattern. High

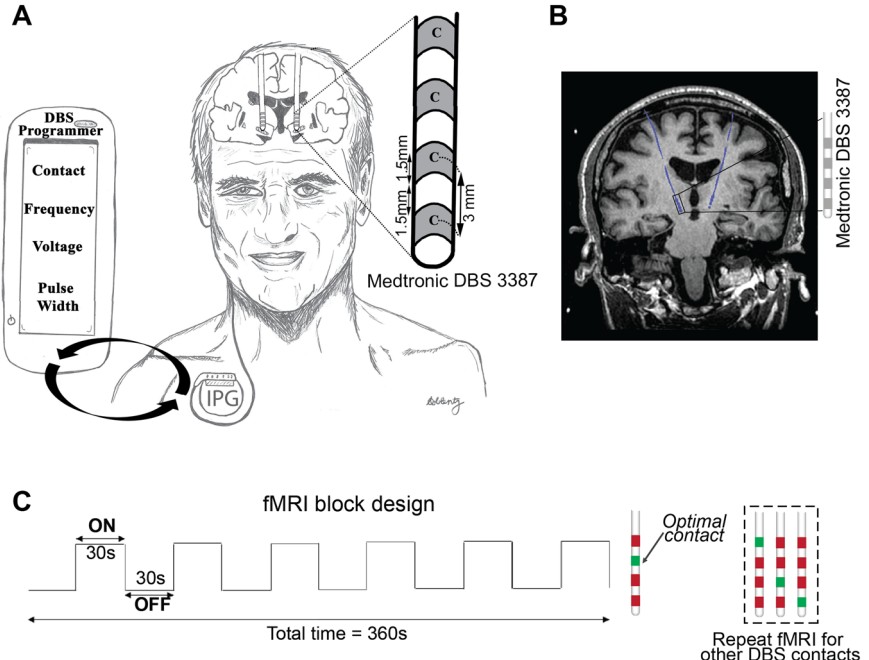

**Fig. 1 Experimental design of 3 T fMRI imaging with DBS activation in PD patients. A** DBS patient implanted with bilateral fully internalized and active DBS electrodes targeting the STN. The DBS lead (Medtronic 3387) has four contacts (width = 1·5 mm) spaced 1.5 mm apart. Using the handheld DBS programmer, DBS programming involves titrating the current delivered by adjusting multiple parameters (i.e., electrode contact, voltage, frequency, and pulse-width) in order to provide the best symptom relief. **B** Coronal T1-weighted image demonstrating a PD patient with fully internalized and active DBS electrodes (blue) implanted in the STN. **C** fMRI block design paradigm used during 3 T fMRI data acquisition. While the patient laid still in the scanner, unilateral (left) DBS stimulation was cycled ON and OFF every 30 s for six cycles. The DBS ON/OFF cycling was manually synchronized to fMRI acquisition. Each fMRI sequence was acquired at either optimal (green) or non-optimal (red) contacts or voltages. In this example, the four contacts were screened with fMRI; the a priori clinically optimal contact (marked in green) and non-optimal contacts (marked in red) are shown. DBS deep brain stimulation, fMRI functional magnetic resonance imaging, PD Parkinson's disease.

(supratherapeutic) voltages produced a relatively stronger BOLD response in the left (ipsilateral) motor cortex and right (contralateral) anterior cerebellum but was also accompanied by increased BOLD signals in non-motor regions such as the inferior frontal and occipital lobes (Fig. 3B). BOLD changes in the right (contralateral) cerebral hemisphere were also seen with high voltages.

Group-level spatial distribution and magnitude of BOLD changes across optimal left unilateral stimulation in the training data (n = 39 patients, Table 1) are shown in Fig. 4. DBS most commonly triggered the largest BOLD increase in the left (ipsilateral) thalamus and BOLD decrease in the left (ipsilateral) motor cortex (Fig. 4A). Due to slight inter-patient electrode location heterogeneity (introduced by subtle but notable differences in brain anatomy and operative lead placement from one patient to the next), conventional group-level (i.e., second-level) fMRI analyses were not optimal for our analysis. Indeed, the individual optimal settings may be considered to engage similar networks while non-optimal settings could recruit different networks depending on electrode position and settings differences across patients. Nevertheless, this type of analysis also showed left (ipsilateral) motor cortex decrease in BOLD signal with optimal stimulation whereas non-optimal stimulation recruited non-motor areas predominantly in the frontal and parietal lobes (Supplementary Fig. 2).

As a preliminary assessment of the effect of a third DBS settings on fMRI patterns, we also performed bilateral DBS stimulation in patients with clinically optimized low (n = 4, 60–80 Hz) and high (n = 6, 150–180 Hz) frequencies in reference to the commonly used 130 Hz (Supplementary Fig. S3). Bilateral stimulation was employed during fMRI to mimic programming

of frequency, in which bilateral electrodes are evaluated simultaneously for clinical efficacy. As the frequency data was acquired with a different paradigm than the contact and voltage data (bilateral, rather than unilateral stimulation), it was not incorporated into the ML model. Similar to optimal contacts and voltages, the motor cortex and thalamus also demonstrated a change in BOLD signal in these patients using low or high optimal frequencies (Supplementary Fig. S3).

To further assess the meaning of the fMRI signal changes with stimulation, we compared changes in BOLD signal when testing stimulation at the electrode contact giving optimal and non-optimal clinical benefits. Across all patients, the location of optimal and non-optimal contacts was not significantly different along the X and Y planes (p > 0.01, two-sided Wilcoxon's rank sum test), however, there was a significant difference in their depth (Z plane) (p = 0.0016, two-sided Wilcoxon's rank sum test) (Supplementary Fig. S4). To permit comparison of the different contacts tested across patients, each patient's optimal contact was defined as the origin (i.e., 0) and non-optimal contacts were mapped by their relative distance (i.e., 3–9 mm) from the optimal contact (Fig. 4B). Patients were grouped on this relative distance scale. The magnitude of the motor circuit BOLD response clearly scaled with stimulation's proximity to the optimal contact. For example, the decrease in BOLD response in the ipsilateral primary motor cortex was significantly greater during stimulation at optimal versus non-optimal contacts (p < 0.05, two-sided Wilcoxon's rank sum test, Fig. 4B, Supplementary Fig. S5A). The contralateral posterior cerebellum also demonstrated significantly greater BOLD signal with stimulation using optimal versus non-optimal contact (p = 0.01, one-sided Wilcoxon's rank sum test) (Supplementary Fig. S6A). This BOLD activation

**Table 1 Demographic information of patients (contact and voltage) included in the machine learning analysis.**

| Patient cohort | Age (mean ± SD) | Gender | Duration of disease (mean ± SD) | UPDRS III (OFF) (mean ± SD) | Levodopa equivalence (mean ± SD) | Days from surgery (mean ± SD) | DBS target | Location of STN electrode (mean MNI coordinates) | Location of GPi electrode (mean MNI coordinates) |
|---|---|---|---|---|---|---|---|---|---|
| Train data—contact (n = 20) | 63 ± 5.7 | 12M 8F | 12.4 ± 4.8 | 34.2 ± 10.4 | 935.6 ± 532.8 | 427.5 ± 488.5 | 18 STN 2 GPi | −13.3, −12.7, −3.6 | −23.2, −5.0, −0.09 |
| Train data—voltage (n = 19) | 61.8 ± 8.4 | 8M 11F | 13.7 ± 5.1 | 38.9 ± 11.3 | 1485.2 ± 662 | 830.8 ± 623.1 | 17 STN 2 GPi | −13.5, −12.6, −3.5 | −20.4, −7.3, −1.5 |
| Test data—preprogrammed (n = 9) | 64.8 ± 9.4 | 8M 1F | 12.4 ± 4 | 28.4 ± 5.3 | 1818.4 ± 1189.1 | 712.1 ± 406.8 | 9 STN | −11.8, −12.7, −5.1 | None |
| Test data—early (n = 9) | 58.5 ± 11.8 | 8M 1F | 9.8 ± 5.4 | 35.8 ± 12.9 | 1450.4 ± 802 | 92.8 ± 94.3 | 8 STN 1 GPi | −12.5, −13.4, −4.4 | −21.2, −7.8, −2.3 |
| Frequency data (n = 10) | 63.2 ± 8.1 | 5M 5F | 13.9 ± 5.8 | 39.4 ± 26.5 | 836.3 ± 407.4 | 1069.36 ± 673.9 | 10 STN | −13.1, −12.2, −3.4 | None |

Patient cohort was divided into three datasets: train, unseen test datasets 1 and 2. Train and unseen dataset 1 include patients a priori clinically optimized. Unseen dataset 2 comprise stimulation naïve patients. Of note, one patient in the unseen dataset 2 (early) underwent the MRI 11.3 months from his surgery, since he had dysphonia after surgery causing a long delay before his programming could be started. Levodopa equivalence was ascertained via the methods of Tomlinson et al.[62]. MNI coordinates of active contacts were obtained using Lead-DBS (Lead-DBS v2.0; https://www.lead-dbs.org/) To obtain mean MNI coordinates of active contacts in the frequency data, right-sided electroides were flipped to the left. As the frequency data were acquired with a different paradigm than the contact and voltage data (bilateral, rather than unilateral stimulation), it was not incorporated into the machine learning model.
UPDRS-III Unified Parkinson's Disease Rating Scale Part III, DBS deep brain stimulation, MNI Montreal Neurological Institute, STN subthalamic nucleus, GPi internal globus pallidus.

pattern was specific to particular regions in the motor circuit; the response in the ipsilateral thalamus, for example, was not significantly different between optimal and non-optimal contact stimulation (Supplementary Figs. S5B, S6A–B, S7A).

We also compared BOLD changes in various brain regions at different voltages. For voltage changes, the BOLD signal in ipsilateral primary motor cortex could not significantly differentiate the optimal from non-optimal voltage settings ($p > 0.05$, two-sided Wilcoxon's rank sum, Fig. 4C) whereas it significantly differentiated the optimal from the subtherapeutic voltage settings in the ipsilateral thalamus ($p = 0.027$, two-sided Wilcoxon's rank sum, Supplementary Fig. S7B). BOLD signal in the ipsilateral pallidum could also distinguish the optimal voltage ($p < 0.05$, two-sided Wilcoxon's rank sum, Supplementary Fig. S6C), indicating that the degree of fMRI response pattern changes across multiple regions—rather than an individual region —is more informative to gauge stimulation efficacy.

**Prediction of optimal contact using ML**. Given our finding of differential fMRI response patterns produced as a function of DBS stimulation site and voltage (Figs. 3 and 4), we next sought to identify the brain regions whose activity would be most informative in predicting clinical benefit. For each patient, normalized mean $t$-values for activation and deactivation were calculated from the statistical response maps for 16 ROIs corresponding to motor areas and areas corresponding to known side-effects of DBS (Fig. 2, see the "Methods" section). Features from these 16 ROIs ($n = 39$, train data, Table 1) were used to derive an ML model that classified a given setting as optimal or non-optimal using linear discriminant analysis (LDA). Frequency data ($n = 10$) were excluded from the ML analysis as it was acquired with bilateral, rather than unilateral left DBS stimulation.

Using a 5-fold cross-validation approach, the combined ML model using both contact and voltage parameter variations achieved 88% training accuracy for classifying optimal versus non-optimal parameter settings ($n = 39$ total, $n = 35$ STN-DBS, and $n = 4$ GPI-DBS; train data, Table 1, Fig. 2, Supplementary Fig. S8). When only motor regions (thalamus, anterior cerebellum, and primary motor cortex) were considered, the training accuracy dropped to 67%, indicating that other regions, including non-motor regions, contribute to optimal contact prediction. Even though the pattern of BOLD signal response associated with optimal GPi-DBS was different than the pattern associated with STN-DBS (Supplementary Fig. S9), when the four GPi-DBS patients were excluded from the train data, the training accuracy decreased from 86% to 81%. This suggests that contributions of BOLD signal patterns from non-optimal GPi stimulation in particular, were beneficial to the algorithm's accuracy.

The model was validated with two additional unseen datasets: a priori clinically optimized and in stimulation naïve patients (Table 1, Fig. 2, Supplementary Fig. S8). For the test dataset of a priori clinically optimized patients ($n = 9$, Table 1), the combined ML model using contact and voltage parameter variations yielded the highest predictive accuracy for optimal settings (Fig. 5A, E) with the lowest false positive rate (Fig. 5A). When only fMRI patterns from contact parameter variations ($n = 20$, training data) were used for training, test accuracy dropped to 63% (Fig. 5C); conversely, training with only voltage parameter variations ($n = 19$, training data) yielded 71% accuracy on the test set (Fig. 5D). To further evaluate predictive validity, we also tested the best-performing classifier (i.e., the combined contact and voltage ML model) on an independent set of stimulation-naïve patients ($n = 9$ total, $n = 8$ STN-DBS and $n = 1$ GPI-DBS; Table 1, Supplementary Fig. S8). This cohort simulates real-

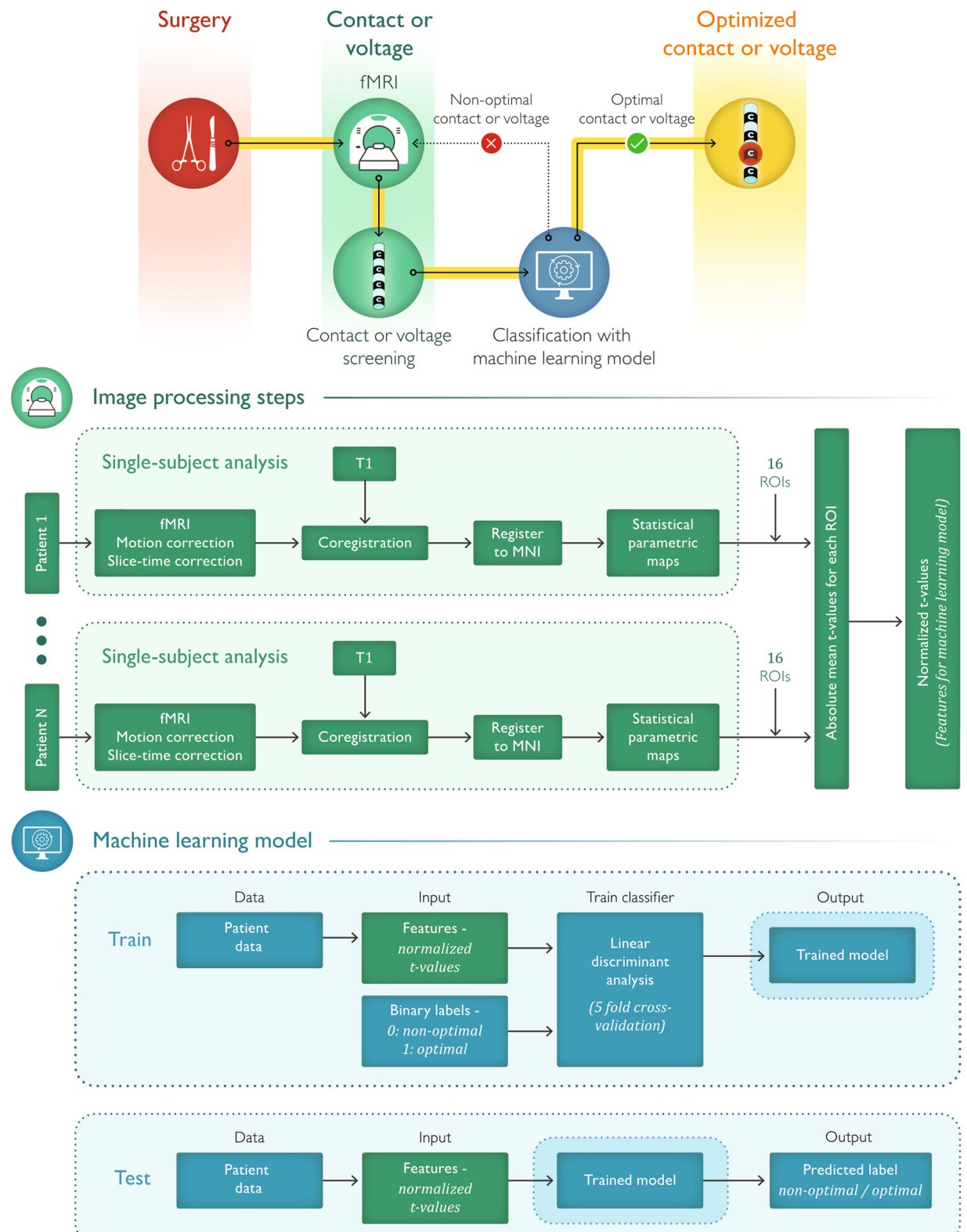

**Fig. 2 Summary of the methods.** (Top row) After DBS surgery, PD patients undergo fMRI with fully implanted and active DBS systems. Contacts or voltages are screened and their associated fMRI patterns are fed into the machine learning model, which classifies the pattern as optimal or non-optimal. (Middle row) Pipeline for fMRI data processing. (Bottom row) Machine learning model is built with a train dataset using linear discriminant analysis and 5-fold cross validation. Then, unseen test datasets can serve as input to the model for validation. fMRI functional magnetic resonance imaging.

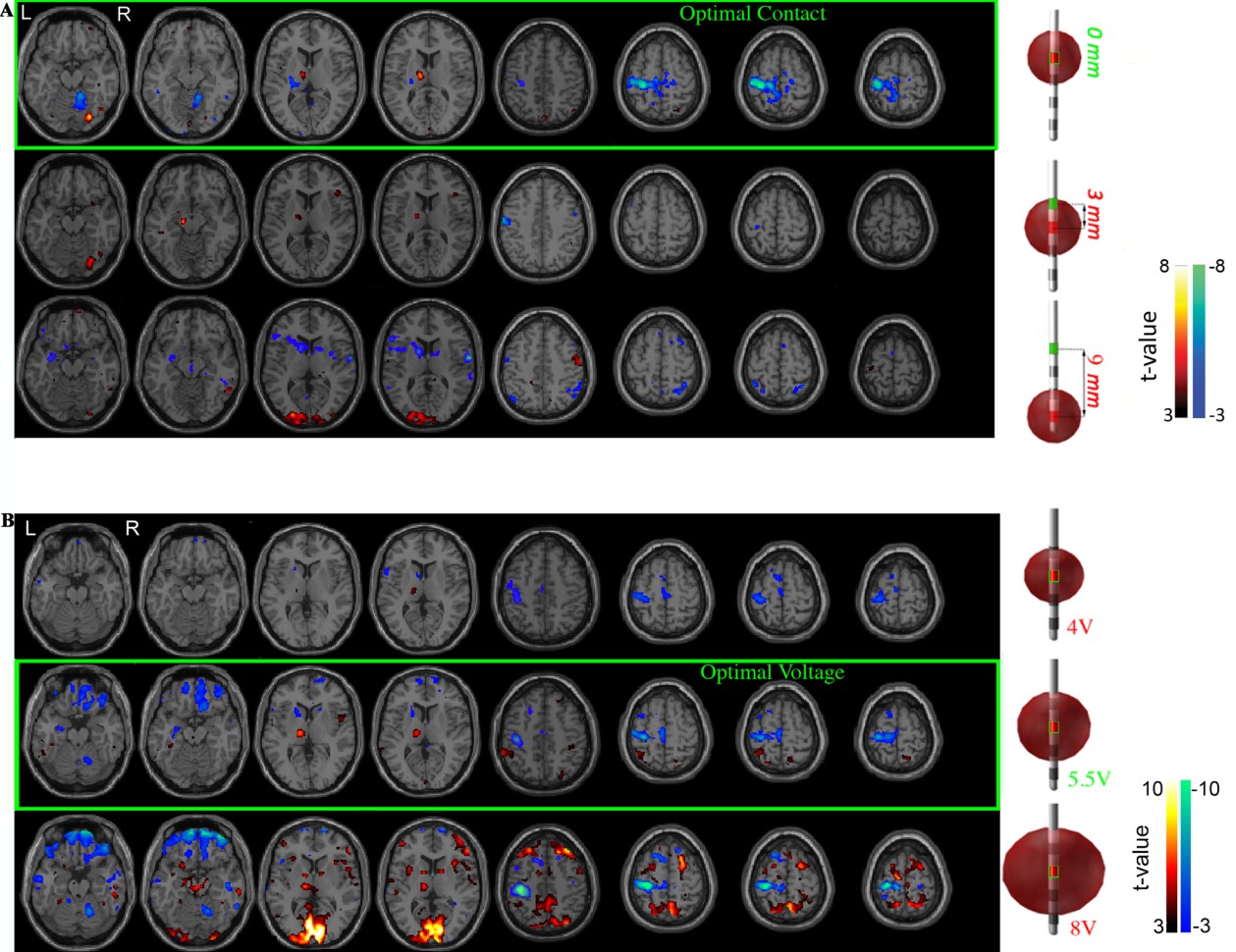

**Fig. 3 Typical pattern of fMRI changes resulting from different settings.** BOLD response maps associated with left DBS-STN stimulation at multiple DBS lead **A** Contacts and **B** voltages for two a priori clinically optimized PD-STN patients. The fMRI BOLD signal changes at the optimal contact (**A** top row) and voltage (**B** middle row) are shown. Brain regions with a significant increase (hot colors, positive $t$-values, DBS-ON > OFF) and decrease (cool colors, negative $t$-value, DBS-ON < OFF) ($p < 0.001$, cluster size $= 50$) in BOLD response were identified. **A** The optimal contact showed changes in BOLD response in the left (ipsilateral) motor cortex and thalamus, and right (contralateral) cerebellum. We considered the clinically optimal contact as the origin (i.e., 0) and the non-optimal contacts were mapped as a function of distance in mm from the optimal contact. **B** When using the optimal stimulation contact, decreasing stimulation amplitude from optimal to low (subtherapeutic) voltage stimulation triggered a decrease in magnitude of the BOLD changes but maintained the topographic pattern. High (supratherapeutic) voltages produced a relatively stronger BOLD response in the left (ipsilateral) motor cortex and right (contralateral) cerebellum but was also accompanied by increased BOLD signal in non-motor regions such as the inferior frontal and occipital lobes. The subtherapeutic voltage was defined as 1.5 V below optimal voltage because a reduction of this magnitude yields a change in clinical status for most PD patients. The supratherapeutic voltage was defined as the voltage just below the side effects threshold (i.e., highest tolerated voltage). BOLD blood-oxygen-level-dependent, DBS deep brain stimulation, fMRI functional magnetic resonance imaging, PD Parkinson's disease, STN subthalamic nucleus.

time programming in patients who have not undergone optimization. These patients subsequently underwent clinical programming by a neurologist blinded to the DBS fMRI results. Here, fMRI features obtained in <30 min of scanning time correctly predicted which DBS settings were deemed clinically optimal—as determined by the neurologist over many programming sessions over ~1–1.5 years—with 76% accuracy (Fig. 5B, E), validating the use of our ML model for patients not yet optimized.

## Discussion
Following the satisfactory resolution of safety concerns in prior studies[7,9,14], we acquired a large cohort of prospective 3 T fMRI in PD-DBS patients and demonstrated a characteristic pattern of brain responses to clinically optimal stimulation. By contrasting these patterns with those obtained during non-optimal

stimulation, we trained and validated a ML model to classify whether a given stimulation setting could be considered clinically optimal in terms of DBS contact and voltage (Fig. 5). For one patient with multiple contact being used clinically (i.e., interleaved, see the "Methods" section), the current binary classifier also predicted multiple contacts as optimal. While there is general agreement that the hot spot for stimulation can be engaged even by a fraction of a contact and that multiple contacts are used to lessen stimulation-induced side effects, it has also been argued that stimulation at multiple levels and with fractions of contact (i.e., directional leads) can provide additional effects such as larger therapeutic windows[21,22]. Given the encouraging results in this relatively small dataset, future classifiers will include more data and incorporate confidence scores for each tested contact to accommodate for less common settings using multiple contacts.

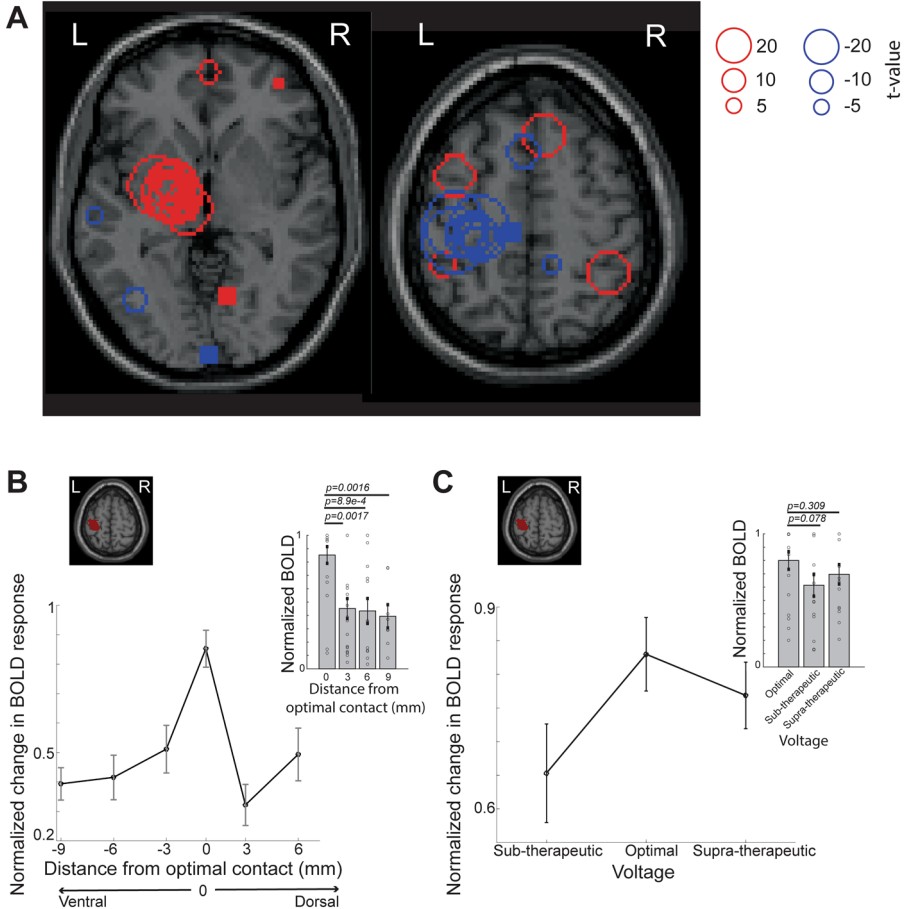

**Fig. 4 Group analysis of fMRI responses to optimal DBS stimulation shows a specific response pattern. A** Distribution of peak *t*-values overlaid on a standard Montreal Neurological Institute (MNI) brain when the clinically optimal left DBS settings are used (*n* = 39 total, *n* = 35 STN-DBS and *n* = 4 GPI-DBS, train data). Red circles reflect increased BOLD activity (DBS ON > OFF) whereas blue circles indicate decreased BOLD activity (DBS ON < OFF). Left thalamic regions showed high overlap of peak activation *t*-values (DBS ON > OFF) across subjects and left motor regions showed peak deactivation *t*-values (DBS ON < OFF) across subjects. **B** The optimal contact was considered the origin (i.e., 0) and the non-optimal contacts were labeled with distances relative to the optimal contact. When the optimal contact was the most dorsal or ventral, the maximum distance to the furthest contact was 9 mm. Changes in BOLD signal in the ipsilateral primary motor cortex in response to stimulation at the optimal and non-optimal contacts on STN-DBS leads are shown. Absolute values of *t*-values at the left primary motor cortex ROI (shaded red) were normalized by *t*-values in the visual and operculum ROIs (*y*-axis). Mean normalized BOLD activity in the left primary motor cortex at the optimal contact was significantly different from the non-optimal contacts 3–9 mm away from optimal location (inset, *n* = 20 (optimal), *n* = 22 (3 mm), *n* = 13 (6 mm), *n* = 8 (9 mm), train data contact with at least one non-optimal contact, Table 1, two-sided Wilcoxon rank sum test). **C** Effects of varying voltage delivered at the optimal contact on BOLD signals are shown. Absolute values of *t*-values at the left primary motor cortex ROI (shaded red) were normalized by *t*-values in the contralateral motor cortex ROIs (*y*-axis). The mean normalized BOLD activity (*t*-values) in the left primary motor cortex (*y*-axis) were maximal at the left optimal contact, but not significantly different from non-optimal voltages BOLD activity (*n* = 19 optimal voltage, *n* = 15 supra-therapeutic, and *n* = 16 sub-therapeutic voltage settings, train data voltage (Table 1), two-sided Wilcoxon's rank sum test). Error bars indicate SEM. Source data are provided as a Source Data file. BOLD blood-oxygen-level-dependent, DBS deep brain stimulation, fMRI functional magnetic resonance imaging, ROI regions-of-interest, STN subthalamic nucleus.

Consistent with previous studies, we found that left unilateral stimulation at the optimal DBS contact or voltage reproducibly engaged the motor circuit[23–26], preferentially modulating BOLD signal in the ipsilateral primary motor cortex, ipsilateral thalamus, and contralateral cerebellum. Engagement of these areas was maintained when stimulation at optimal contact, voltage, or frequency was applied. A prominent finding was the decrease in BOLD signal in the primary motor cortex seen with STN-DBS. In line with our observations, several PET studies have reported reduced cerebral blood flow in the primary and premotor cortices during STN-DBS[27,28]. Moreover, STN-DBS has been suggested to decrease pathological beta oscillations in the primary motor cortex[29]. Still, other imaging studies have reported no changes[30] or increased cerebral blood flow in the motor cortex[16] during STN-DBS. Reasons for these discrepancies may include (1)

different imaging modalities (i.e., MRI vs. PET), (2) the time of scan data acquisition after surgery, and/or (3) resting state versus task-based acquisition. Similar to our findings, thalamic, and cerebellar brain activity changes have been reported upon acute STN stimulation[11,15,26,31]. It is unclear whether the thalamic BOLD signal changes are secondary to afferent/efferent thalamic activation or contiguous spread from the STN stimulation target, especially if the most dorsal contact was used. The mechanism of action of the striking deactivation effect on the primary motor cortex is not clear but could include a primary retrograde effect on the motor cortex mediated via the hyperdirect pathway or an anterograde effect mediated through direct and indirect basal ganglia circuitry[32]. The prominent cerebellar effect is likely a transynaptic circuit effect secondary to the changes in motor cortex activity with STN DBS.

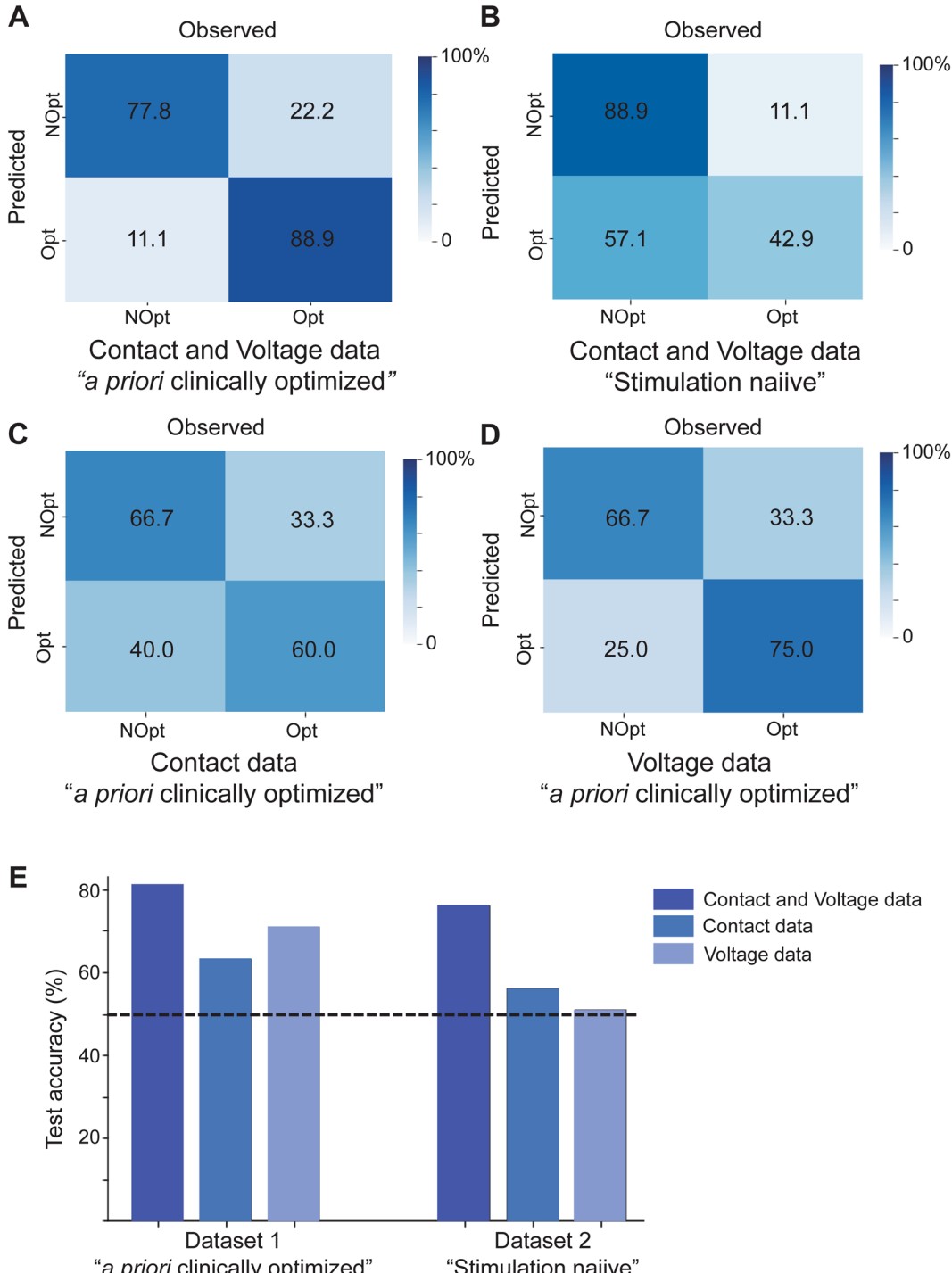

**Fig. 5 fMRI responses predict optimal DBS parameters.** Confusion matrices depicting the performance of classifiers trained to identify optimal DBS settings using features from **A** contact and voltage cohorts, **C** contact cohort alone, and **D** voltage cohort alone in an independent test set (n = 9 a priori clinically optimized patients). **B** Confusion matrix depicting the performance of the classifier trained to identify optimal DBS settings using features from contact and voltage cohorts in an independent test set (n = 9 stimulation naïve patients). **E** Summary of performance (overall accuracy) for classifiers in **A–D**. Bars from dataset 1 depict classifier test accuracy on n = 9 a priori clinically optimized patients. Bars from dataset 2 depict classifier test accuracy on n = 9 stimulation naïve patients. Dashed line indicates chance at 50% accuracy. Source data are provided as a Source Data file. DBS deep brain stimulation, fMRI functional magnetic resonance imaging, NOpt non-optimal, Opt optimal.

BOLD response maps associated with non-optimal stimulation showed engagement of non-motor circuits, including the visual cortices and operculum. The mechanisms underlying these effects are not fully understood, but there are at least two possible explanations: (1) STN is surrounded by several white matter tracts[33]; (2) STN has associative and limbic territories beyond its dorsolateral motor regions[34]. As the stimulated area moves further from dorsolateral STN, off-target STN areas and tracts are likely stimulated. This undesired recruitment of non-motor areas could be considered "friendly fire" and be responsible for some of

the adverse side-effects (e.g., muscle contractions, oculomotor dysfunction, slurred speech, cognitive, psychiatric, and gait disturbances) commonly observed with PD-DBS[35].

Commonalities in neuroimaging findings suggest that clinical benefits from various PD therapies are underpinned by a partially overlapping neuroanatomical network. For example, levodopa administration has also been shown to decrease primary motor cortex activity[36,37]. Yet another study comparing metabolic changes with STN-DBS and levodopa pharmacotherapy found that both treatments increased activity in SMA and decreased activity in the primary motor cortex. However, differences were seen: increased putaminal metabolism was only seen with levodopa whereas prefrontal areas showed increased metabolism with STN-DBS and decreased metabolism with levodopa[38]. At the network level, both DBS and levodopa appear to restore the abnormal Parkinson's disease-related spatial covariance pattern (PDRP)[18,39]. It is also interesting that GPi-DBS studies was also shown to normalize PDRP[40].

DBS physicians often need numerous hours of clinical testing, stretched over multiple hospital visits, to discern the optimal contact. Even in PD patients, for whom immediate clinical feedback is present, patients obtain largest clinical benefits after ~1 year of programming. This clinical assessment is especially challenging when symptoms are mild (e.g., in patients without rigidity) and/or when patients possess diminished communicative faculties. At our institution, patients are initially followed up weekly for one month with visits that may last 1–2 h[4]. Then, patients are scheduled for monthly appointments (~1 h in duration) for the first year. The visits eventually become yearly to provide continuous monitoring and adjustments. In the US, a single programming visit is estimated to cost over 1000 USD[41]. Additionally, the advent of more electrode contacts in newer DBS leads, as in for example directional leads, introduces even more programming possibilities and complexity. Time constraints and patient fatigue make it impracticable to thoroughly assess a large number of stimulation parameters via clinical means; this restriction could possibly be mitigated by the fMRI-based method presented here. Given the present results, it is conceivable that both contact and voltage settings could be efficiently optimized using an fMRI-based workflow (Supplementary Fig. 10). In this theoretical scenario, patients would undergo fMRI screening to identify the optimal electrode contacts followed by voltage adjustments with further adjustments taking place as required. Optimizing the contact (4 fMRI image acquisitions (1 per contact) each lasting 6.5 min) and voltage (3 fMRI image acquisitions (high, low, and intermediate voltages) each lasting 6.5 min) could be done within a 1 h-long fMRI, requiring an MRI technologist and a staff able to change DBS programming. We propose that the fMRI-guided programming tool is an objective and individualized measure of clinical benefits in PD patients with STN-DBS. This in turn may streamline current DBS programming, with the possibility of increasing the clinical benefits for PD patients. Importantly, this fMRI tool provides direct insight into brain responses to stimulation and does not rely on assumptions such as estimation of volume of tissue activated (VTA)[42]. Prospective trials comparing the utility and accuracy of this fMRI-based programming tool with traditional empirical programming would be the logical next step. Parameters to track might include clinical benefits, time to stimulation optimization, number of hospital visits, and cost-effectiveness analysis. Finally, it is conceivable that this proof-of-concept method can be extrapolated to both other DBS parameters, such as frequency and pulse width, as well as to other DBS indications, in particular conditions without much immediate clinical response to inform programming.

There are a number of limitations to our findings. The order in which fMRI data were acquired order was not entirely randomized (i.e., the clinically determined optimal contact was usually assessed first). Furthermore, while we would have ideally provided washout time between fMRI series, we chose to keep scanning sessions as short as possible in light of the frailty inherent to this patient population. However, the short total MRI time required for our predictive model may be of value considering MRI availability and cost as well as patient tolerability. The brain network responses to changes in frequency, pulse width, and stimulation polarity remain to be examined. While DBS Medtronic model 3387 (1.5 mm between DBS contacts) is routinely used at our institution, future studies incorporating other DBS hardware could build a more generalizable and robust predictive model. Similarly, the recent designation of DBS systems as full-body eligible (and the use of body-transmit coils) with 1.5 T MRI could enable more widespread use of this method to map brain responses to DBS. While the use of 3 T MRI is desirable for increased signal-to-noise ratio, we have also shown that DBS at 1.5 T could also yield satisfactory fMRI data[43]. Further, DBS in PD patients leads to an immediate change in clinical state (e.g., decreased rigidity or tremor), introducing a possible confound in BOLD signal interpretation. Changes in motor cortex, thalamic, and cerebellar brain activity in DBS patients with asymptomatic physical states at rest (i.e., anesthetized PD and essential tremor patients[11,44]) as well as animal models[45] suggest that our fMRI pattern is a direct DBS-driven effect and not a consequence of clinical improvement. The changes in the BOLD signal could also be due the normalization of abnormal brain metabolism in PD patients[18]. Despite previous studies reporting variability in the HRF across brain regions and across individuals, we used the same canonical HRF (double gamma function) to model the BOLD signals across all brain regions and patients. Using different HRF would likely improve the sensitivity of our data. Future analysis will include HRF determination as a part of the analysis workflow. There are also limitations related to the predictive modeling we chose. Typically, generalizable ML models require large pools of data to avoid overfitting. Even with the current data limitations in our study, we were able to achieve >80% sensitivity for optimal parameter prediction indicating the promise of fMRI-feature driven ML models for DBS parameter optimization. The ML model used features from selected ROIs based on previous literature and our experience (see the "Methods" section) but it did not exhaustively sample the entire brain. Typically, in neuroimaging data, the number of features are much greater than the number of independent samples, leading frequently to model overfitting. Given limited patient data, we overcame the overfitting by restricting the number of ROIs included. With larger data samples, classifiers trained with features from finer brain parcellations incorporating dimensionality reduction techniques (like principal component analysis, recursive feature elimination[46], etc.) would lead to more robust predictive models. We also did not investigate whether the optimal contact could also be predicted with stimulation location. However, the significant difference in the location of optimal and non-optimal contacts in our study and the published relationship between stimulation of "sweet spots" and clinical outcomes[47] suggest the potential utility of using stimulation location in future applications. Preceding the fMRI, the contact closest to the "sweet spot" could be first determined, which could then guide the contact screening with fMRI and potentially reduce the number of sessions needed. Notwithstanding these limitations, the high predictive accuracy of our ML model, coupled with the fact that DBS surgeries were performed by three neurosurgeons and DBS optimization by two neurologists increases the generalizability of our findings.

Previous DBS fMRI studies have made a number of interesting observations including (1) acute changes in brain activity with

stimulation[10,11,15,26,31,48], (2) connectivity changes associated with beneficial stimulation[13,30,49–53], or (3) restoring brain connectivity based on stimulation overlap with DBS targets[12]. In contrast to the work presented here, most of these studies have used bipolar stimulation[11,12,53] and have been conducted at lower field strengths (1.5 T)[10–13,26,31,48,50,53]. To our knowledge, the current study is nearly three times larger than any prospective fMRI DBS study to date. We have shown that high-quality prospective fMRI data can be translated into a potentially clinically useful tool. The fMRI acquisitions we present here were done with omnidirectional electrode contacts and open loop stimulation based on careful safety testing[7,9,14]. As new electrodes and stimulation technologies emerge and become more widespread, including for example directional electrodes and closed loop DBS systems, the attendant safety and the impact on functional imaging with stimulation using these systems will also need to be evaluated.

In conclusion, we present reproducible functional maps of therapeutic DBS activity in the largest prospective cohort of patients, derived using fMRI, as an objective clinical tool for DBS programming. DBS patients attend numerous costly and lengthy physician visits in order to repeatedly titrate stimulation parameters in pursuit of an optimal clinical result. Lack of immediate clinical feedback in response to stimulation in non-PD DBS patients (e.g., dystonia or depression), makes DBS programming particularly challenging. Notably, poor programming has been suggested as a possible contributing reason for the failure of randomized control trials in DBS for depression[54]. Our results show that fMRI can rapidly define the optimal DBS stimulation in PD patients. Obtaining DBS-induced fMRI brain signatures associated with optimal clinical benefits, will not only allow us to obtain a better understanding of the mechanism of action of DBS, but could also facilitate individualized medicine for our patients and may represent a step towards the possibility of autonomous, closed-loop DBS programming.

## Methods

**Participants.** Following institutional research ethics board approval (University Health Network, 14-8255), PD patients who had previously undergone DBS surgery targeting STN or GPi at Toronto Western Hospital were enrolled in this study as a part of an ongoing observational clinical trial (Table 1, Supplementary Fig. S11, $n = 67$, Age $= 62.9 \pm 8$, 41 males, 26 females; #NCT03153670, Responsible party: Andres M. Lozano, University Health Network, Toronto). GPi is also a commonly targeted structure in the management of PD[1]. Although both sites arguably provide similar motor benefits, there are differences: STN contributes to medication intake reduction whereas GPi may be better suited for PD patients with cognitive impairment and medication-associated dyskinesias[55,56]. Yet, the afferent and efferent circuitry for each target are different[57]. To assess whether different PD-DBS targets could also contribute to the ML model, we explored the ML model training accuracy with and without GPi-DBS patients ($n = 4$) (Supplementary Fig. S8). While we recruited all patients within these inclusion criteria, patients were invited to volunteer for the study and it is plausible that they may have displayed similar characteristics, for example in terms of personality and inclination to participate in trials.

DBS surgeries were performed by three neurosurgeons (A.M.L., S.K.K., M.H.). The inclusion criteria were (1) participants receiving active STN- or GPi-DBS, (2) ability to provide written informed consent, and (3) specific models of Medtronic DBS hardware, including DBS leads (3387, 28 cm; Medtronic, Minneapolis, MN), extension wire (37086, 60 cm; Medtronic, Minneapolis, MN) and IPG (Activa PC 37601, Activa RC 37612, Medtronic, Minneapolis, MN). Participants undergoing 3 T MRI were also required to have DBS hardware geometry similar to previous phantoms[7]. The optimal fMRI data (3 T and body-transmit coil) were able to be acquired outside vendor guidelines[58] based on our unique experience with 3 T MRI in DBS patients[7,9,14]. As recommended, a member of the clinical team was present to monitor patients during the MRI session. Prior to MRI scanning, informed consent for participation in the study was obtained. The protocols were approved by the Research Ethics Board at the University Health Network. The current study was further approved by the local ethics committee of the University Health Network in accordance with the Declaration of Helsinki. On average, fMRI data were acquired 18.4 months [1.4–73.3 months] after DBS surgery (Table 1).

All the patients that were recruited in this observational trial are reported in this manuscript and had reached the endpoint time (i.e. optimal clinical programming after surgery) to assess for fMRI brain changes with DBS (Supplementary Fig. S11). Forty (out of the 67) patients included in the current study were included in a prior study, which described the safety profile of 3 T MRI and DBS-associated artifact on fMRI sequences[14]. The present data was used to investigate fMRI brain changes associated with contacts and voltages. Data of future enrolled patients will investigate changes with other DBS programming settings (i.e. frequencies and pulse widths).

**Study design.** We primarily recruited PD patients with STN-DBS (Table 1). To assess the specificity of our results to STN, a small number of GPi-DBS PD patients were also included (Table 1). Since contact and voltage selection are usually the first DBS parameters to be assessed during post-operative programming, in this study fMRI patterns of brain activation at the a priori clinically determined optimal contacts or voltages were contrasted with those of non-optimal contacts or voltages. As a preliminary assessment of the effect of a third DBS settings on fMRI patterns, we also performed bilateral DBS stimulation in patients with clinically optimized low ($n = 4$, 60–80 Hz) and high ($n = 6$, 150–180 Hz) frequencies in reference to the commonly used 130 Hz (Supplementary Fig. S3). Bilateral stimulation was employed during fMRI to mimic programming of frequency, in which bilateral electrodes are evaluated simultaneously for clinical efficacy. As the frequency data was acquired with a different paradigm than the contact and voltage data (bilateral, rather than unilateral stimulation), it was not incorporated into the ML model.

A 3 T MRI (GE HDx, Milwaukee, WI) and either a transmit-receive head coil (GE Model 2376114) or a body-transmit coil (GE 2380637-2) were utilized to acquire 6.5-min fMRI sessions using a 30 s DBS-ON/OFF cycling paradigm (Supplementary Table 2, Fig. 1). To control for any potentially confounding signal produced by PD medications, all patients were instructed to take their final medication dose the night preceding MRI acquisition. Shortly before initiating MRI scanning, the DBS system was turned off, with localizer and structural images being acquired without stimulation prior to fMRI. Hence, at the start of fMRI acquisition, the DBS system had been turned off for ~15 min. fMRI was acquired in 67 PD patients for a total of 203 fMRI sessions. Fifty-nine of these patients had been receiving chronic DBS stimulation and their stimulation at the time of the fMRI (or the settings at 1 year after the surgery for those who underwent the fMRI before 1-year post-op) were deemed clinically optimized (Table 1). Clinically optimal DBS settings for these patients were obtained using published algorithms[4,5]. Nine patients were stimulation-naïve patients who recently (<1 month of programming) underwent DBS surgery (i.e., no clinically defined optimized DBS settings at the time of the MRI, Table 1). They received clinical programming by a neurologist blinded to the fMRI results. Two movement disorder neurologists, who previously published programming algorithms[4,5], were involved in the optimization of the patients.

During fMRI acquisition, all patients were set to a 30 s DBS-ON/30 s DBS-OFF cycling paradigm (Fig. 1). The cycling was manually synchronized to the fMRI acquisition period. fMRI sequences were acquired using either different contacts along the DBS electrode or different voltages. Contacts or voltages were a priori categorized as optimal or non-optimal by a movement disorder neurologist for the previously programmed patients (Table 1). Conversely, optimal settings were not known at the time of the fMRI in the stimulation naïve patients, who subsequently received clinical programming by a neurologist blinded to the fMRI results. During the fMRI, the patients were blinded to the DBS settings.

Unilateral left DBS stimulation was delivered during fMRI acquisition of contact and voltage data. As reported previously[15], this was done to differentiate the unilateral and contralateral BOLD signal changes, as well as to attempt to mimic DBS programming, which usually entails evaluating one electrode at a time. The order in which non-optimal contact or voltage stimulation was delivered was randomized. For frequency data, bilateral stimulation was performed as DBS programming for frequency is commonly evaluated using both electrodes simultaneously.

Most DBS patients included in the study had a monopolar electrical configuration (43/57 monopolar patients (Supplementary Table 3). Although only bipolar stimulation during MRI acquisition is approved by the vendor guidelines[58], we specifically used the native stimulation settings (including monopolar stimulation) because we have shown that conversion methods to bipolar stimulation yield inconsistent fMRI patterns[9]. We have also shown that the fMRI pattern of brain changes are largely reproducible[9]. Other programming parameters (frequency and pulse-width) were kept constant throughout the fMRI acquisition period to mimic the programming process during which contacts and voltages are usually assessed first.

For patients in whom we tested different contacts, the highest tolerated voltage was used when they could not tolerate the clinically prescribed optimal voltage at a non-optimal contact. Most DBS patients in whom we tested different contacts had a monopolar electrical configuration. Non-monopolar configurations (i.e., bipolar, double monopolar, or interleaved, Supplementary Table 3) use more than one DBS contact to deliver stimulation. For the few patients programmed with non-monopolar configurations in whom we tested different contacts, we recorded optimal stimulation using non-monopolar settings as clinically determined by their programming neurologist. The remaining contacts were considered non-optimal and tested individually as monopolar configurations. For the patients in whom we tested different voltages, both low (subtherapeutic) and high (supratherapeutic)

voltages were delivered. The subtherapeutic voltage was defined as 1.5 V below optimal voltage because this decrease will yield a change in clinical status for most PD patients. The supratherapeutic voltage was defined as the voltage immediately below the side effects threshold (i.e., highest tolerated voltage).

As detailed in a previous study[14], a complete neurological exam was performed following scan completion and the acquired MR images were immediately reviewed to detect any acute intracranial changes. In addition, the impedances of the DBS contacts were recorded before and after scanning to assess for alterations in electrical circuit integrity and peri-electrode tissue changes (e.g., edema and hemorrhage).

**fMRI analysis.** Exploratory fMRI analyses were performed to establish a reproducible fMRI pattern of brain activation by contrasting optimal and non-optimal contacts and optimal and non-optimal voltages. Then, these fMRI brain response patterns were used to build an ML model capable of predicting the optimal, patient-specific contact setting (Fig. 2). The ML model was trained on 39 a priori clinically optimized patients ($n = 35$ STN-DBS and $n = 4$ GPi-DBS) and subsequently tested on two unseen datasets: nine a priori clinically optimized patients (Table 1) and nine stimulation-naïve patients who had recently undergone surgery (Table 1).

**Single subject analysis.** All fMRI data were slice time corrected, motion corrected, rigidly registered to a T1-weighted image, non-linearly registered to a standard space MNI brain, and spatially smoothed using a FWHM 6 mm gaussian kernel in SPM12 (http://www.fil.ion.ucl.ac.uk) (Fig. 2) and MATLAB (Mathworks, Natick, MA, USA). To account for artifacts due to head motion in PD patients, we used the Art toolbox (https://www.nitrc.org/projects/artifact_detect)[59] to detect and remove volumes with motion >1.5 mm. Overall, for any given patient, this resulted in the removal of a maximum of 6 volumes (3.3%) from the total volumes acquired. The estimated parameters for 6-degrees of motion were used as regressors to the design matrix used for calculating statistical parametric maps. To ascertain that the observed changes were not related to head-motion related to DBS stimulation paradigm, we correlated 6-degrees of motion parameters with DBS ON/OFF block design. There was no significant correlation between motion parameters and DBS ON/OFF block design (Supplementary Fig. 12). Statistical parametric maps (functional response t-maps) were estimated using a 30 s DBS-ON/OFF block design with the canonical double gamma function for modeling the hemodynamic response function (HRF). The absolute t-values (BOLD changes) were normalized by mean positive t-values in areas presumed to be involved in non-optimal stimulation. For contact patients, absolute t-values were normalized by the t-value in the visual cortex and operculum ROIs whereas for voltage patients, the right motor cortex t-values were used for normalization. These areas for normalization were chosen based on initial data exploration: non-optimal contact stimulation tended to recruit side effects areas such as the visual cortex and supratherapeutic voltage triggered contralateral brain changes in our data. Functional response maps were corrected for multiple comparisons using a p-value of 0.001, with cluster level thresholding of 50 voxels, to give an overall p-value of <0.05 for visualization. While the cluster threshold is used for visualization purposes, the ML model was constructed using unthresholded t-values to retain the full spectrum of the data.

**Group-level analysis.** Given small but notable inter-patient deviations in electrode contact location, which were attributable to subtle disparities in both patient-specific anatomy and technical electrode placement, conventional fMRI second-level analysis was not optimal. We favored exploration of the differences in fMRI activity changes between optimal and non-optimal settings in the train data ($n = 39$) and frequency data ($n = 10$) by showing the spatial distribution and magnitude of BOLD changes. However, for completeness, we also performed conventional fMRI second-level analysis on the train data ($n = 39$ total, $n = 35$ STN-DBS and $n = 4$ GPi-DBS). A general linear model (GLM) was applied to the contrast maps of normalized t-values from each subject at the second level. For patients in whom we tested different contacts, BOLD changes as a function of distance from the optimal contact were then assessed (Fig. 4B, Supplementary Fig. 7A). The optimal contact was considered the origin (i.e., 0) and the non-optimal contacts were labeled with distances relative to the optimal contact. For patients in whom we tested different voltages, BOLD changes as a function of voltage were assessed (Fig. 4C, Supplementary Fig. 7B).

**Hemodynamic response function (HRF) estimation.** BOLD signal is presumed to follow a predictable response over time and can be represented by a function called the HRF. The HRF represents the BOLD signal fluctuation over time and can be approximated with different models. Despite previous studies reporting variability in the HRF across brain regions and across individuals[60], we used the same canonical HRF (double gamma function) to model the BOLD signals across all brain regions and patients. To validate the use of the canonical HRF, we estimated the HRF for DBS and compared it to the canonical double gamma function. The canonical HRF used in the analysis was found to be similar across multiple brain areas including the primary motor cortex, in which the observed BOLD signal was significantly correlated with the canonical HRF ($r = -0.7$) (Supplementary

Fig. S13). Because of this similarity between the canonical double gamma function and the HRF, we used this function across all brain regions.

**ML model.** PET and SPECT studies, and to a lesser extent fMRI experiments, in DBS patients have informed the region-based analysis. Prospective fMRI studies in DBS patients remain few and far between due to safety concerns. PET and SPECT conducted in PD patients have confirmed distributed motor circuit engagement for STN and GPi. Acute changes with stimulation consistently engaged the motor hubs of the CSTC circuit including the precentral gyrus, thalamus, STN, and to a lesser extent the pallidum, supplementary motor area, and cerebellum[16–19]. Other regions such as the operculum and visual cortex were included to account for speech issues and visual disturbances experienced with non-optimal settings. Thus, to perform a region-based analysis for each patient, average t-values were determined for 16 ROIs including the thalamus, pallidum, primary motor cortex, anterior cerebellum, and supplementary motor area. These were derived from a functional atlas[61] (Fig. 2). ROIs included regions in the thalamic-motor circuit such as the thalamus, pallidum, primary motor cortex, anterior cerebellum, and supplementary motor area. Additionally, ROIs from other areas that could be related to common adverse effects (e.g., speech and gait disturbances) observed in PD-DBS patients at non-optimal contacts and voltages during our MRI sessions were included in the analysis. As a result, primary and secondary visual cortex, operculum, and posterior cerebellum were also included. This resulted in 32 features for each contact or voltage tested (16 positive and 16 negative t-values corresponding to increase and decrease of BOLD signal from 16 ROIs). The mean t-values were then normalized by mean positive t-values in the visual cortex and operculum ROIs (contact patients) and right (contralateral) motor cortex (voltage patients). This was done to compare t-values of each ROI across patients and to account for the adverse effects given that the aim of DBS tuning is to maximize motor benefits while minimizing adverse effects.

**ML model: training.** Using 39 patients a priori clinical optimized ($n = 39$ total, $n = 35$ STN-DBS and $n = 4$ GPi-DBS; train data, Table 1, Supplementary Fig. S8), normalized mean t-values extracted from 16 ROIs (32 features for each patient) were used to classify the optimal and non-optimal contacts and voltages using a linear-discriminant analysis (LDA) within a 5-fold cross validation framework (MATLAB, Mathworks, Natick, MA, USA). Note that a priori clinical optimized patients were randomly assigned to the train and test datasets.

To explore the robustness of our model, additional train datasets were considered (Supplementary Fig. S8). First, to investigate the importance of other non-motor and sensory regions, the model was trained only with features from thalamus, motor and cerebellum ROIs ($n = 39$ patients). This resulted in 9 ROIs (18 features) instead of 16 ROIs described above. Then, only patients with contact parameter variations ($n = 20$, Table 1) or only patients with voltage parameter variations ($n = 19$, Table 1) were used to train the model. Further, to ascertain the influence of GPi-DBS patients on the model, a model was trained excluding the 4 GPi-DBS patients. As the frequency data were acquired with a different paradigm than the contact and voltage data (bilateral, rather than unilateral stimulation), it was not incorporated into the ML model.

**ML model: testing.** The validity of the LDA ML classifier model was assessed by testing it on two groups of unseen, independent data sets ($n = 18$, Table 1, Supplementary Fig. S8: 9 patients a priori clinically optimized by the neurologist ($n = 9$ STN-DBS) and 9 ($n = 8$ STN-DBS and $n = 1$ GPi-DBS) stimulation-naïve patients prior to initiation of programming. Each patient's fMRI response maps were fed to the ML model to prospectively predict the optimal DBS setting (Fig. 2).

**Reporting summary.** Further information on research design is available in the Nature Research Reporting Summary linked to this article.

## Data availability
The fMRI datasets analyzed are not publicly available due to data privacy regulations of patient data. Upon reasonable request, the study protocol and individual de-identified participants' raw fMRI data will be available to investigators from the corresponding author using private online cloud storage for reproducibility analyses. The analyzed fMRI data used for Fig. 3A and B and Supplementary Fig. S2 are publicly available at Github (https://github.com/radhika-madhavan/fMRI-DBS) and Zenodo (https://doi.org/10.5281/zenodo.4633710). Source data are provided with this paper.

## Code availability
The custom code that supports the central findings of this study are publicly available at Github (https://github.com/radhika-madhavan/fMRI-DBS) and Zenodo (https://doi.org/10.5281/zenodo.4633710).

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

## Acknowledgements

We acknowledge Dr. Ileana Hancu and Eric Fiveland from GE Global Research for their help with the MRI protocol, safety testing, and providing the fMRI-DBS synchronization box. We would like to thank our illustrator, Andrew O'Connor, for helping with the design of Fig. 2 and Supplementary Fig. S10. We would also like to thank GE Global Research and the Michael J. Fox Foundation for their financial contribution. The supporting party (GE Global Research) contributed to the study design, data acquisition, analysis, interpretation, and writing of the manuscript. The Michael J. Fox Foundation did not contribute to the study design, data acquisition, analysis, interpretation, or writing of the manuscript. The corresponding author confirms that he had full access to all the data in the study and had final responsibility for the decision to submit for publication.

## Author contributions

A.B. and R.M.: experimental design and set up, data acquisition, analysis, data interpretation, manuscript writing. S.E.J. and M.R.: experimental design and set up, data acquisition, data interpretation, manuscript edits. D.X., A.L., J.G., B.L. and S.P.: experimental design and set up, data acquisition, manuscript edits. S.K.K., M.H., A.F., V.P., A.C., G.J.B.E., R.G., R.P.M. and W.K.: experimental design, data acquisition, data interpretation, manuscript edits. J.A.: experimental design and manuscript edits. A.M.L.: experimental design, data acquisition, data interpretation, manuscript edits.

## Competing interests

R.M., S.E.J. and J.A. are employees at General Electric. B.L. is now a Medtronic employee; he was not at the time of this work completion. Medtronic had no role in data acquisition, analysis, or interpretation. A.F. serves as a consultant for Medtronic, Abbott, Boston Scientific, Brainlab, Ceregate, and Medtronic, he received research grants, personal fees and non-financial support from Boston Scientific, Brainlab and Medtronic and personal fees from Abbott and Ceregate, all outside the submitted work. S.K.K. reports honorarium and consulting fees from Medtronic. A.M.L. serves as a consultant for Medtronic, Abbott, Boston Scientific, and Functional Neuromodulation. A.B., R.M., S.E.J., J.A. and A.M.L. have intellectual property related to this manuscript. The other authors report no conflict of interest concerning the materials or methods used in this study or the findings specified in this paper.
