## [Peer Review File · Nature Communications]

Editorial Note: This manuscript has been previously reviewed at another journal that is not operating a transparent peer review scheme. This document only contains reviewer comments and rebuttal letters for versions considered at Nature Communications. Mentions of prior referee reports have been redacted.

Reviewer #1 (Remarks to the Author):

Authors have addressed all issues raised in the previous round of reviews (in a different journal). I have no further comments.

In my view this paper could truly become a corner-stone for a new clinical avenue. What if – in the future – DBS programming indeed would take place in the MRI? It is somewhat surprising that this hasn't been explored earlier (also little has been done along similar lines with EEG). Likely reason for fMRI is that scanning patients with electrodes is not straight-forward – here the Toronto group has made a lot of experience before carrying out this study. However, both DBS centers and electrode manufacturers alike would likely turn their head to this study and explore their options. Despite MRI being costly, the cost pales in comparison to tedious DBS programming procedures that often take >6 months and many man-hours to complete.

Reviewer #2 (Remarks to the Author):

The authors have addressed the specific concerns raised by the earlier manuscript. While this is ultimately an interesting proof-of concept study, it is still far removed from clinical needs. Currently, efforts are made to predict DBS settings through recordings of beta-oscillations. Potential safety issues attendant to the directional electrodes will also need to be evaluated. Even so, the manuscript is fascinating and can be considered for publication in the journal.

Reviewer #3 (Remarks to the Author):

Review comments: NCOMMS-20-32931-T

[Redacted]

Knowing the story from the 1st submission I was enthusiastic to re-read this revised work as I strongly believe that this work has great potential for changing the field of DBS programming. However, I noticed some issues shortly after looking at the Result section. In fig 3A, the authors show three fMRI maps and claim that the optimal contact is the deepest (contact # 0, bottom row) while the figure legend states "top row" is the optimal location representing contact #3 (this happens to us all). More confusing is that the authors point to Supplementary Fig. S1 where they attempt to visualize in 3D the electrode and contact location of the same patient and there they show contact #3 (top contact; at least based on the shown VTA). And here is where it's really getting interesting, I went back and looked at my comments from the 1st review and there, FOR THE EXACT SAME PATIENT (identical fMRI maps), the Optimal contact was reported to be the dorsal contact (C#3). In fig. 3B where they compare the maps for optimal voltage, once again, for the exact same fMRI maps, now the active contact is C#1 (2nd from bottom) while in fig. 3 in the [Redacted] version it shows up as contact #2 (3rd from bottom). So, for the same patient/data, two conflicting results are being presented.

Group level analysis shown in Supplementary Fig. S2:

- a) please show the same cortical slices across the motor region that a fair evaluation of the amplitude changes can be done.
- b) panel D of Supp. Fig.2, at supra-threshold voltage, the group analysis map shows only minor BOLD cortical changes while in Fig. 3, bottom row, there are significant changes, perhaps even larger than for the optimal voltage.

Reviewers' Comments:

Reviewer #1 (Remarks to the Author):

Authors have addressed all issues raised in the previous round of reviews (in a different journal). I have no further comments.

In my view this paper could truly become a corner-stone for a new clinical avenue. What if – in the future – DBS programming indeed would take place in the MRI? It is somewhat surprising that this hasn't been explored earlier (also little has been done along similar lines with EEG). Likely reason for fMRI is that scanning patients with electrodes is not straight-forward – here the Toronto group has made a lot of experience before carrying out this study. However, both DBS centers and electrode manufacturers alike would likely turn their head to this study and explore their options. Despite MRI being costly, the cost pales in comparison to tedious DBS programming procedures that often take >6 months and many man-hours to complete.

- Thank you for the careful review of our manuscript and for recognizing the potential impact of our work becoming a “corner-stone” for a new clinical avenue.

Reviewer #2 (Remarks to the Author):

The authors have addressed the specific concerns raised by the earlier manuscript. While this is ultimately an interesting proof-of concept study, it is still far removed from clinical needs. Currently, efforts are made to predict DBS settings through recordings of beta-oscillations. Potential safety issues attendant to the directional electrodes will also need to be evaluated. Even so, the manuscript is fascinating and can be considered for publication in the journal.

- Thank you for the careful review of our manuscript and for agreeing that we have addressed the specific concerns. We also thank you for saying our work is fascinating and can be considered for publication.
- We have incorporated details regarding potential safety issues attendant to directional electrodes in the text of the discussion

The fMRI acquisitions we present here were done with omnidirectional electrode contacts and open loop stimulation based on careful safety testing.^{7,9,14} As new electrodes and stimulation technologies emerge and their use become more widespread, including for example directional electrodes and closed loop DBS systems, the attendant safety and the impact on functional imaging with stimulation using these systems will also need to be evaluated (Discussion page 18).

Reviewer #3 (Remarks to the Author):

Review comments: NCOMMS-20-32931-T

[Redacted]

Knowing the story from the 1st submission I was enthusiastic to re-read this revised work as I strongly believe that this work has great potential for changing the field of DBS programming.

- Thank you for the careful review of our manuscript and for the kind comments.

However, I noticed some issues shortly after looking at the Result section.

In fig 3A, the authors show three fMRI maps and claim that the optimal contact is the deepest (contact # 0, bottom row) while the figure legend states “top row” is the optimal location representing contact #3 (this happens to us all).

- Thank you for pointing out this discrepancy and the confusion that it generates. It arises as a consequence of a labeling error we made in the figure. In attempting to revise Figure 3 for NCOMMS-20-32931-T, we switched the rows without correctly addressing the VTA/electrode schematic on the right of the figure. The figure should show that the top (i.e. dorsal) contact is the optimal contact (as can be seen in Figure 3 in the original [Redacted] version). The corrected figure 3 attached below has the appropriate labelling. We apologize for the mislabeling in the previously revised version and the confusion this caused.

More confusing is that the authors point to Supplementary Fig. S1 where they attempt to visualize in 3D the electrode and contact location of the same patient and there they show contact #3 (top contact; at least based on the shown VTA).

- The Supplemental Figure in question S1A, B is correct, while Fig. 3A in the revised version had the error which has now been corrected as described above.

And here is where it's really getting interesting, I went back and looked at my comments from the 1st review and there, FOR THE EXACT SAME PATIENT (identical fMRI maps), the Optimal contact was reported to be the dorsal contact (C#3).

- As noted above, the optimal contact as shown in the [Redacted] version is correct (i.e. top/dorsal contact).

In fig. 3B where they compare the maps for optimal voltage, once again, for the exact same fMRI maps, now the active contact is C#1 (2nd from bottom) while in fig. 3 in the [Redacted] version it shows up as contact #2 (3rd from bottom). So, for the same patient/data, two conflicting results are being presented.

- Again, this was a mistake that occurred during the revision process. The active contact in Fig. 3B should be contact #2 (3rd from bottom), as it is in the [Redacted] version. In addition, we noted that Supplemental Fig. 1C, D (which is a 3D visualisation of the active contact in Fig. 3B) is also incorrect. As with Fig. 3B, the active contact should be contact #2 (3rd from bottom). The updated corrected figures are incorporated below.

We have corrected these labelling errors in the figures below:

Figure 3: Typical pattern of fMRI changes resulting from different settings. (A) BOLD response maps associated with left DBS-STN stimulation at multiple contacts along the DBS lead for an *a priori* clinically optimized PD-STN patient. The fMRI BOLD signal changes at the optimal contact (top row) and non-optimal contacts (middle and bottom rows) are shown. Brain regions with a significant increase (hot colors, positive t-values, DBS-ON>OFF) and decrease (cool colors, negative t-value, DBS-ON<OFF) ($p<0.001$, cluster size=50) in BOLD response were identified. We considered the clinically optimal contact as the origin (i.e., 0) and the non-optimal contacts were mapped as a function of distance in mm from the optimal contact. The optimal contact showed change in BOLD response to be in the left (ipsilateral) motor cortex and thalamus, and right (contralateral) cerebellum. (B) BOLD response maps associated with left DBS-STN stimulation at multiple voltage settings for another *a priori* clinically optimized PD-STN patient. The figure shows the fMRI BOLD signal change at the optimal voltage (middle row) as well as at subtherapeutic (top row) and suprathreshold (bottom row) doses. The subtherapeutic voltage was defined as 1.5 volt below optimal voltage because a reduction of this magnitude will yield a change in clinical status for most PD patients. The suprathreshold voltage was defined as the voltage just below the side effects threshold (i.e., highest tolerated voltage). Hot and cool colors indicate significant positive and negative t-values (described in A).

Supplementary Fig. S1: Electrode localization for patients in Figure 3. 3-D reconstruction of the electrodes for the contact (A, B) and voltage (C, D) patients in Figure 3. Their volume of tissue activated (shaded red) are overlaid on sagittal (A, C) and coronal (B, D) T-1 weighted standard brain (MNI space ICBM 2009b NLIN asymmetric). The subthalamic nucleus is shown in orange (Ewert et al., 2018). ICBM = International Consortium of Brain Mapping; MNI = Montreal Neurological Institute

Group level analysis shown in Supplementary Fig. S2:

a) please show the same cortical slices across the motor region that a fair evaluation of the amplitude changes can be done.

- Supplementary Fig. S2 has been modified and now shows cortical slices at the same level in the motor region.

Supplementary Fig. S2: Group analysis of fMRI. (A) Group-level analysis of the fMRI response when the left optimal DBS was turned ON ($n=39$ total, $n=35$ STN-DBS and $n=4$ GPI-DBS, train data) showed significant BOLD changes in motor areas. (B) Non-optimal left DBS contacts ($n=50$ fMRI sessions from 20 patients, train data) stimulation showed a similar response pattern to (A) but with a reduced magnitude. In addition, fMRI analysis of the non-optimal left contacts showed significant decreased BOLD response in the operculum and visual areas. Brain regions with significant increase (hot colors, positive t-values, DBS-ON>OFF) and decrease (cool colors, negative t-value, DBS-ON<OFF) ($p<0.001$, cluster size=50) in BOLD response were identified. (C) Non-optimal sub-therapeutic voltages (voltage < optimal voltage) showed responses in the motor cortex but with reduced amplitude, while (D) suprathreshold voltages (voltage > optimal voltage) showed activation in non-motor regions. BOLD = blood-oxygen-level-dependent; DBS = deep brain stimulation; fMRI = functional magnetic resonance imaging; GPi = internal globus pallidus; STN = subthalamic nucleus; Left = left; R = right.

b) panel D of Supp. Fig.2, at supra-threshold voltage, the group analysis map shows only minor BOLD cortical changes while in Fig. 3, bottom row, there are significant changes, perhaps even larger than for the optimal voltage.

- Thank you for this comment. As mentioned in the text, we think that the “conventional” fMRI group-level analysis is not optimal for our data. This is due to the slight inter-patient electrode location heterogeneity (introduced by subtle but notable differences in brain anatomy and operative lead placement from one patient to the next), which likely introduces different network activations/deactivations for individual non-optimal settings. This could explain why the group level results do not match the single example shown in Figure 3B. We adjusted the results to emphasize this point.

Due to slight inter-patient electrode location heterogeneity (introduced by subtle but notable differences in brain anatomy and operative lead placement from one patient to the next), conventional group-level (i.e., second-level) fMRI analyses were not optimal for our analysis. Indeed, the individual optimal settings may be considered to engage similar networks while non-optimal settings could recruit different networks depending on electrode position and settings differences across patients. Nevertheless, this type of analysis also showed left (ipsilateral) motor cortex decrease in BOLD signal with optimal stimulation whereas non-optimal stimulation recruited non-motor areas predominantly in the frontal and parietal lobes (Supplementary Fig. 2) (Results page 8)

Tables and Figures summary of changes to accommodate the comments:

- **Table 1:** No change.
- **Figure 1:** No change.
- **Figure 2:** Minor formatting changes.
- **Figure 3:** Modified in light of reviewer comments.
- **Figure 4:** No change.
- **Figure 5:** No change.
- **Supplementary Table 1:** No change.
- **Supplementary Table 2:** No change.
- **Supplementary Table 3:** No change.
- **Supplementary Fig. S1:** Modified in light of reviewer comments.
- **Supplementary Fig. S2:** Modified in light of reviewer comments.
- **Supplementary Fig. S3:** No change.
- **Supplementary Fig. S4:** No change.
- **Supplementary Fig. S5:** No change.
- **Supplementary Fig. S6:** No change.
- **Supplementary Fig. S7:** No change.
- **Supplementary Fig. S8:** No change.
- **Supplementary Fig. S9:** No change.
- **Supplementary Fig. S10:** No change.
- **Supplementary Fig. S11:** No change.
- **Supplementary Fig. S12:** No change.
- **Supplementary Fig. S13:** No change.

Reviewer #3 (Remarks to the Author):

The authors have addressed the issues raised in the previous reviews